# Exploring Effective Stimulus Encoding via Vision System Modeling for Visual Prostheses

**Chuanqing Wang**[1]*, **Di Wu**[1]*, **Chaoming Fang**[1], **Jie Yang**[1]†, **Mohamad Sawan**[12]†

[1] CenBRAIN Neurotech, School of engineering, Westlake University, Hangzhou.
[2] Westlake Institute for Optoelectronics, Fuyang, Hangzhou, 311421, China.
`{wangchuanqing, wudi, yangjie, sawan}@westlake.edu.cn`

## Abstract

Visual prostheses are potential devices to restore vision for blind people, which highly depends on the quality of stimulation patterns of the implanted electrode array. However, existing processing frameworks prioritize the generation of stimulation while disregarding the potential impact of restoration effects and fail to assess the quality of the generated stimulation properly. In this paper, we propose for the first time an end-to-end visual prosthesis framework (**StimuSEE**) that generates stimulation patterns with proper quality verification using V1 neuron spike patterns as supervision. **StimuSEE** consists of a retinal network to predict the stimulation pattern, a phosphene model, and a primary vision system network (PVS-net) to simulate the signal processing from the retina to the visual cortex and predict the firing rate of V1 neurons. Experimental results show that the predicted stimulation shares similar patterns to the original scenes, whose different stimulus amplitudes contribute to a similar firing rate with normal cells. Numerically, the predicted firing rate and the recorded response of normal neurons achieve a Pearson correlation coefficient of 0.78.

## 1 Introduction

Visual prostheses are potential devices to restore vision for patients with severe vision disorders, such as age-related macular degeneration (AMD) and retinitis pigmentosa (RP) (Busskamp et al., 2010; Cehajic-Kapetanovic et al., 2022; Prévot et al., 2020). These prostheses utilize an image sensor to record external scenes and a processing framework to predict stimulus for the electrode array (Soltan et al., 2018). The implanted electrode array stimulates ganglion cells with predicted stimulus to restore visual perception in the visual cortex (Berry et al., 2017; Sahel et al., 2021). In the whole workflow, the processing framework to obtain suitable stimulation patterns is the most critical factor in affecting visual prostheses' restoration effect, attracting many research endeavors (Turner et al., 2019). The performance of stimulation patterns predicted by the processing framework is highly related to the choice of the supervised signal and processing model (Turner et al., 2019).

One strand of the processing framework to construct stimulation patterns utilized a region-based detection method to process the recorded external frame signals (Figure 1(a)). The region of interest of the input frame was converted to grayscale and directly mapped to the electrode location for stimulation without any supervised signal (Guo et al., 2018; Wang et al., 2023). Due to the unsupervised nature of these methods, the quality of the stimulation patterns can not be verified before deploying this framework in implanted devices for animal experiments.

Another strand of research adopted neural networks to predict stimulation patterns and a biological phosphene model to simulate phosphene and generate decoded images (Figure 1(b)) (Burcu et al., 2022; Granley et al., 2022; Wang et al., 2022a). The phosphene represents visual perception without light entering the eyes. Then, the original image is used as a supervised signal to update the stimulation pattern, making the output decoded image similar to the original one. However, we argue that the visual perception in the visual cortex is neural spikes but not images. Consequently, this processing

---

*These authors contribute equally to this work.
†Corresponding author.

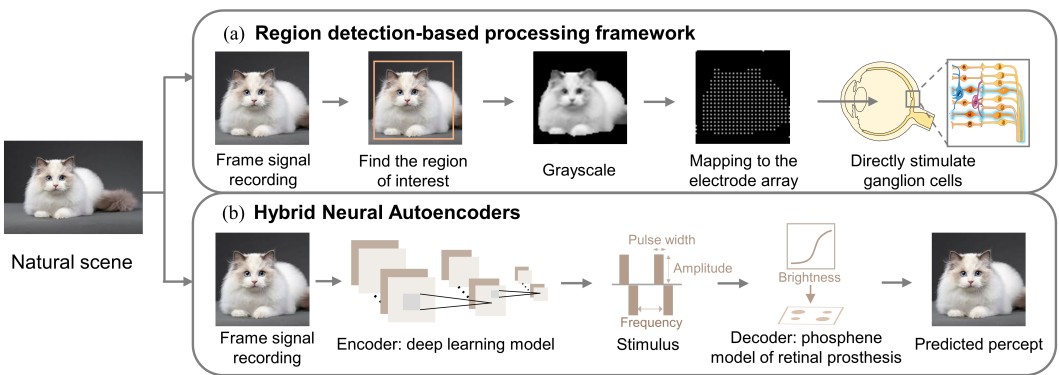

Figure 1: Illustration of two existing processing frameworks. (a) The external scene is recorded with a series of frame signals in a region detection-based processing framework. Such a framework utilizes region-based detection to detect a region of interest and simplifies it with grayscale and pixelation techniques to obtain stimulation patterns. These patterns are mapped into an electrode array to stimulate ganglion cells directly for vision rehabilitation. (b) Hybrid neural autoencoders (usually deep neural networks) take frame signals as input and predict suitable stimuli for retinal prosthesis. Another decoding model is used to simulate phosphene and generate the decoding image. The original image is adopted as a supervision signal during autoencoder training, whose performance is evaluated by comparing the similarity between the decoded image and the original image.

framework only finds suitable stimulation patterns that favor original image reconstruction rather than finding suitable patterns for better vision perception in the cortex. Thus, this stimulus with low biological similarity restricts the vision restoration effect of visual prostheses at a low-level (Montazeri et al., 2019).

To remedy the inadequacy of existing frameworks, the proposed framework (StimuSEE) generates stimulation patterns that trigger the same spike responses in the visual cortex as normal visual perception when given the same scene information. As demonstrated in Fig. 2, StimuSEE combines a retinal network, a phosphene model, and a primary visual system network (PVS-net). Retinal prostheses require the processing framework to be energy efficient for prolonged use. To ensure energy efficiency, the retina network adopts a spiking recurrent neural network (SRNN) to predict stimulation patterns for electrode arrays. Compared to the traditional CNN models, SRNNs have a low spike firing rate during their calculation process without the need for floating point multiplication. The phosphene model and PVS-net are designed specifically to validate the effectiveness of predicted stimulation patterns. The phosphene model is a biological simulation model to accurately simulate phosphene in the retina after stimulation by the implanted prosthesis. PVS-net mimics the function from the retina to the visual cortex and predicts the firing rate of primary visual cortex (V1) neurons. The predicted values of StimuSEE fit well with that of normal neurons, whose performance achieves 0.78, evaluated by the Pearson correlation coefficient. We hope that the resemblance shown between the predicted stimulation patterns and neural spikes with normal neurons could contribute to a more practical visual prosthesis development.

In short, we summarize the main contributions of our work as the following:

- To the best of our knowledge, we are the first to propose an end-to-end visual prosthesis framework (**StimuSEE**) that generates stimulation patterns with proper quality verification using V1 neuron spike patterns.

- We designed a phosphene model and a PVS-net to simulate the visual signal process from the retina to the primary visual cortex to validate the effectiveness of predicted stimulation patterns.

- Beside quality verification of stimuli, **StimuSEE** is designed in an energy-efficient manner aiming to bring truly practical visual prostheses into reality.

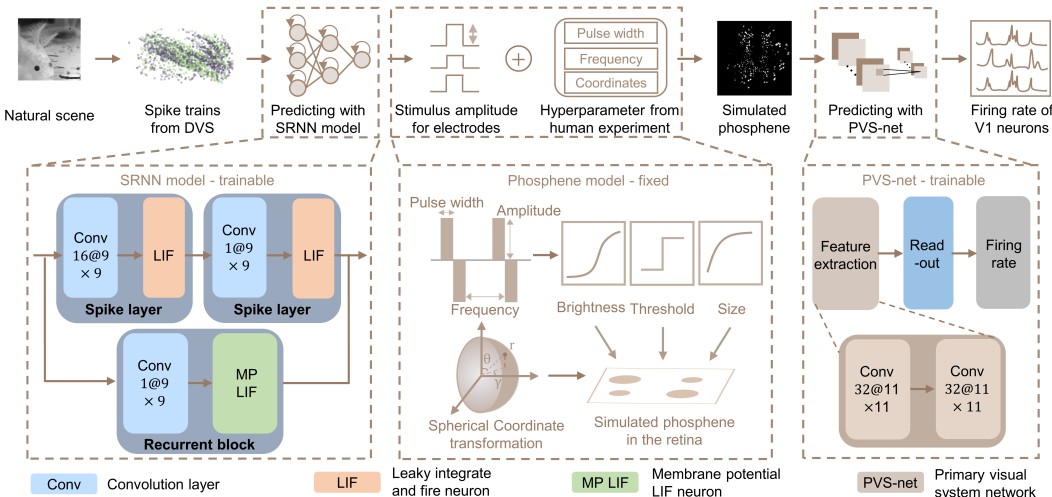

Figure 2: The proposed optimization framework mainly comprises the retina network, phosphene model, and PVS-net. The retina network utilizes dynamic vision sensors (DVS) to encode natural scenes into sparse spike trains. Then, the SRNN model is designed to predict the stimulus amplitude of the electrode array with input spikes. This model consists of two spike layers and one recurrent block. The phosphene model integrating predicted stimulus amplitude and hyperparameters from human experiments to simulate phosphene in the retina. This model fully considers the impact of spherical coordinate transformation, phosphene brightness, threshold, and size effects. The PVS-net can mimic the function of the visual system, especially lateral geniculate nucleus (LGN) and primary visual cortex, with the input of simulated phosphene. The model output is the firing rate of multiple V1 cells compared with the recorded response of normal cells to evaluate stimulus performance.

## 2 RELATED WORKS

**Region-based detection.** The initial processing framework for visual prostheses adopted region-based detection to find a region of interest first and simplify it to obtain stimulation patterns (Li et al., 2018; Itti et al., 1998). A pixelated model was proposed to detect the interest region by selecting attention-grabbing local features, then pixelated this region to stimulation pattern (Li et al., 2005; Boyle et al., 2008). Later, a self-adaptive FCM clustering method was proposed to identify all clusters containing objects and choose the one closest to the center as a region of interest (ROI). This ROI was transformed into a grayscale image, and edge detection was used to extract edges from background information (Wang et al., 2016; Parikh et al., 2013). The final pattern was constructed with grayscale foreground region and background edge information. This method preserves more features for recognition compared with other region-based detection methods. In addition, several studies focused on constructing stimulation patterns from dynamic scenes (Guo et al., 2019; Zhou et al., 2003). A combined processing strategy, including edge detection and zooming techniques, was proposed to segment moving objects from dynamic scenes (Guo et al., 2018; Wang et al., 2014; Barnich & Van Droogenbroeck, 2011).

**Hybrid neural autoencoder with phosphene model.** Apart from predicting stimulation patterns, several studies focused on constructing a biological simulation model to simulate phosphene and verify the quality of generate patterns (van der Grinten et al., 2022). A stimulation vector mapping method is proposed to convert stimulation patterns into light spots in the electrode location (van Steveninck et al., 2022). Then, a biologically plausible simulation model was designed to consider several biological features, including phosphene brightness and size (Granley & Beyeler, 2021). Recently, a hybrid neural autoencoder was proposed to predict the stimulation patterns with the CNN model and generate simulated phosphene with the above phosphene simulation model (Granley et al., 2022). This phosphene was decoded to original images to verify its performance. Another processing framework utilized a similar method to generate simulated phosphene but added a reinforcement learning agent to improve the performance in decoding original images (Burcu et al., 2022).

## 3  STIMUSEE

In this study, we employ the neural firing patterns of V1 neurons to assess the efficacy of stimulation. Our fundamental premise is straightforward: If the neural spikes collected in V1 neurons of visually impaired individuals following stimulation resemble the spikes recorded from V1 neurons in sighted individuals, it suggests an effective stimulation for vision restoration. StimuSEE comprises a retina network, a phosphene model, and a PVS-net (Figure 2). The retina network predicts the stimulation patterns that are applied to the stimulation array of the implanted prosthesis. The retina network adopts the SRNN model and takes input from a dynamic vision sensor to mimic the function of the human retina for high-quality stimulus while ensuring low power consumption. The phosphene model fully considers the coordinate system transition and phosphene brightness, size, and threshold effect to simulate phosphene at the retina accurately. Finally, we adopt a PVS-net to mimic the function from the retina to the primary visual cortex (V1) and predict the firing rate of multiple V1 neurons (Lindsey et al., 2019). The predicted neurons' firing rate from the PVS-net is then compared with the recorded response of normal V1 neurons to evaluate the performance of the generated stimulation.

### 3.1  RETINA NETWORK

Inspired by the spike-based processing characteristic of the human retina, we proposed a retina network that includes a dynamic vision sensor (DVS) and SRNN model to predict stimulation patterns with spike signals (Figure 2). Recent studies demonstrated that spike-based perception and processing methods with DVS and SRNN models achieve better prediction performance with much less energy consumption than frame-based sensor and CNN model (Wang et al., 2022a). The dynamic vision sensor could encode scenes into sparse spike trains. The SRNN model comprises two spike layers and one recurrent block, suitable for processing the input spatial-temporal signal.

We utilize a dynamic vision sensor and a synchronous recording system to encode scene information into sparse spike trains. This vision sensor only encodes the moving part of scenes into spike trains, which are of high temporal resolution and low data volume. Then, the recording system receives and stores these spike trains synchronously through TCP/IP protocol. Before constructing an event dataset for the proposed SRNN model, it is necessary to denoise the recorded spike trains to improve their quality (Leñero-Bardallo et al., 2011). Finally, due to the large spatial resolution of the vision sensor, the input resolution of spike trains is shrunk to $120 \times 120$ for the SRNN model.

The proposed SRNN model is designed to predict stimulation amplitude with the input spike trains (Figure 2). This model utilizes spike layers and a recurrent block to achieve spatial-temporal feature extraction in a low power consumption condition. Both spike layers and recurrent blocks use the convolution layer to extract spatial features. In addition, the temporal feature extraction ability depends on the dynamic characteristics of the Leaky Integrate and Fire (LIF) and Membrane Potential LIF (MP_LIF) neurons. For LIF neurons, the membrane potential update value is obtained by input spikes multiplied by corresponding weights (Figure 3(a)). Then, the new membrane potential must be compared with the threshold to determine whether to fire a spike. The neuron will fire a spike when membrane potential exceeds the threshold (Figure 3(b)). Thus, LIF neurons could memorize the temporal feature with updated membrane potential.

For MP_LIF neuron, the calculation process and membrane potential updated details of MP_LIF neuron are depicted in Figure 3(c, d). The dynamic characteristic of MP_LIF can be written as:

$$V_t = (1 - \frac{1}{\tau})V_{t-1} + \frac{1}{\tau}X_t, \tag{1}$$

where the membrane time constant $\tau$ controls the balance between remembering $X_t$ and forgetting $V_{t-1}$, which is similar to the function of the recurrent layer (Zhu et al., 2022). Thus, it can be considered a simple version of the recurrent layer. The SRNN model with spatial-temporal feature extraction ability is suitable for processing input dynamic spike signals and can predict the stimulation amplitude for retinal prostheses.

### 3.2  PHOSPHENE MODEL

The phosphene model is designed to simulate phosphene in the retina according to the stimulus of the electrode array, which includes stimulation amplitude, duration, and frequency (Figure. 2). The

Figure 3: The dynamic features of LIF and MP_LIF neurons. (a, b) The updated membrane potential is obtained by adding corresponding weight of input spikes. This potential is added to existing membrane potential of LIF neuron, which is used to compared with threshold to decide whether to fire a spike. (c, d) The MP_LIF neuron has same membrane potential update mechanism. However, at each time step, it outputs the real-valued membrane potential rather than spikes.

amplitude comes from the results of the above SRNN model, and the pulse width and frequency are hyperparameters summarized from human experiment (van der Grinten et al., 2022). Phosphene means visual perception without light entering the eyes. The feature of phosphene is recorded in patients with retinal implants. After the stimulation of each electrode, the patients will report the location, size, and brightness of the phosphene. These features are summarized as a dataset to obtain suitable parameters for the phosphene model. This model first maps the stimulation parameters with the location of the electrode array. Then, it converts the location in one hemisphere to the plane coordinate system and calculates the phosphene brightness and size with the stimulus parameters. Finally, whether to generate a phosphene is also decided by the threshold of retinal ganglion cells.

**Coordinate system transition**   The implanted electrode array is located at the surface of the retina hemisphere, which utilizes spherical coordinates to represent the location information. For calculating the specific location of phosphene, it is necessary to convert Spherical coordinates to Cartesian coordinates. We adopt a reverse wedge-dipole visuotopic model to achieve this conversion (Polimeni et al., 2006).

$$z = \Lambda^{-1}\left(\frac{ab(e^{\frac{w}{k}} - 1)}{b - ae^{\frac{w}{k}}}\right), \tag{2}$$

where $z$ and $w$ are the locations of the visual field and one hemisphere. $k$ is a scaling factor that scales the mapping to realistic proportions in retina distance, and $a$ and $b$ are parameters that control the singularities of the dipole model.

**Phosphene brightness, size, and threshold**   The phosphene brightness is affected by various factors, including the stimulation parameters, the distance to the stimulation electrode, and the distance to the soma along the curved axon. To calculate the effective stimulation intensity, the remaining current amplitude is multiplied by the frequency and duration.

$$I_{eff} = max(0, (I_{stim} - I_{leak}) \cdot P_w \cdot f), \tag{3}$$

where $I_{leak}$, $P_w$, and $f$ are leak current, pulse width, and frequency. The phosphene is determined by the effective stimulation intensity:

$$F_{bright} = \begin{cases} I_{eff} & \text{if } I_{eff} \geq \eta \\ 0 & \text{if } I_{eff} < \eta \end{cases}, \tag{4}$$

where $\eta$ is the threshold to control the intensity. The phosphene size $F_{size}$ is only affected by the stimulation amplitude but not with duration or frequency (van der Grinten et al., 2022). Then, $F_{streak}$ modeling phosphene elongation is a function of time (Granley & Beyeler, 2021). More details are summarized as follows:

$$F_{size} = a_1 I_{stim} + b_1, \tag{5}$$
$$F_{streak} = -a_2 t^\lambda + b_2, \tag{6}$$

where $a_1, b_1, a_2, b_1, \lambda$ are hyperparameters to modify the function to fit the recorded users' response. The final phosphene brightness is contributed by three parts: efficient stimulation amplitude ($F_{bright}$), the distance to the electrode (radial decay rate, $\rho$), and the distance to the soma (axonal decay rate, $\lambda$).

$$Brigheness = \max_{\mathbf{a} \in A} \sum_{e \in E} F_{\text{bright}} \exp\left(\frac{-d_e^2}{2\rho^2 F_{\text{size}}} + \frac{-d_s^2}{2\gamma^2 F_{\text{streak}}}\right), \tag{7}$$

where $A$ is the cells' axon trajectory, $E$ is the set of electrodes, $d_e$ is the path to the electrode, and $d_s$ is the path length along the axon trajectory (Jansonius et al., 2009). In sum, the above phosphene model is designed to simulate phosphene in the retina after stimulated by a retinal prosthesis, taking coordinate system transition and threshold function into consideration compared with other phosphene models, making it more biologically plausible (Granley et al., 2022).

**Parameter selection**    The hyperparameters of the phosphene model utilized in this work are based on various experimental and modeling results from different components of corresponding research: In Eq. 2, we use $a = 0.75$, $b = 120$, $k = 1.73$, based on a fitting method of (Polimeni et al., 2006) on data of an human experiment from (Horton & Hoyt, 1991). In Eq. 3, Eq. 4, $Pw = 3.3ms$, $f = 30Hz$, $threshold = 0.3mW/mm^2$ are obtained from the setting and human experimental results with retinal prosthesis (Granley & Beyeler, 2021). We use $I_{leak} = 23.9\mu A$ based on a data fitting from (van der Grinten et al., 2022). In Eq. 5, Eq. 6, $a_1$, $a_2$, $b_1$, and $b_2$, $\lambda$ is adopted from a retinal phosphene model (Granley et al., 2022). In Eq. 7, $d_s$, $d_e$, $\rho$, and $\gamma$ was obtained from a human retina model (Jansonius et al., 2009).

**Surrogate method for phosphene model**    The phosphene model is a complex function during simulating phosphenes, affected by many parameters and experiment setups. However, in this proposed processing framework, we reduced the entire model to a function of the input stimulus and carefully determined the appropriate values for these parameters. In addition, the threshold function that decides whether to generate phosphene is not differentiable, hindering the backpropagation process. Thus, we adopt an adjusted ReLU function $f(x) = ln(1 + e^{(x-\eta)})$ to replace threshold function to make it differentiable in the training, where $\eta$ is the threshold in Eq. 4.

### 3.3    Primary visual system network

The primary visual system network (PVS-net) is adopted to predict the firing rate of V1 neurons with the simulated phosphene. This network comes from a unified theory of early visual representations from the retina to cortex (Lindsey et al., 2019; Tanaka et al., 2019). This theory utilizes a retina-net and a vision system network to explore visual information processing features and find biological similarities between the proposed network and visual processing features of human beings. Thus, we adopt PVS-net to replace the function of the lateral geniculate nucleus and primary visual cortex. We utilize an online trainable feature bank to capture semantic aware neural responses upon different visual stimulation. In particular, the feature bank contains 50 embeddings $\{e_i\}_{i=1}^{50}$ with the same shape as the output $x$ of the feature extraction module of the PVS-net. During the fitting process of the spike number of V1 neurons, we first replace $x$ with the closest embedding $e_k$ in the feature bank:

$$\mathcal{R}(x) = e_k, \ where \ k = argmin_j \|x - e_j\|_2, \tag{8}$$

where $\mathcal{R}(\cdot)$ denotes the replacement operation. We then apply a read-out layer to predict the firing rate of V1 neurons:

$$f(x) = ReLU( FC(\mathcal{R}(x))) \tag{9}$$

where $f(x)$, $FC$, $ReLU$ means the firing rate of V1 neurons, the fully connected layer, and the activation function (Cadena et al., 2019).

## 4    Experiments

### 4.1    Experimental settings

**Dataset.**    In this paper, we validate the proposed StimuSEE framework with the DVS-V1 dataset (Wang et al., 2022b; Cadena et al., 2019). The dataset comprises 7250 visual stimuli with corresponding V1 responses. The original visual stimuli are selected from the Imagenet and transformed into sparse spike trains by a DVS sensor. We also designed a synchronous recording system to record these spike signals in high quality. More details about this recording system and the quality of the spike signal are summarized in the appendix. The V1 responses were recorded from macaque monkeys with a 32-channel microelectrode array when displaying the above-selected images to these monkeys. Because this experiment was conducted for four sessions, each stimulus has four different V1 responses. The average value of these responses is used as labels to train the proposed framework.

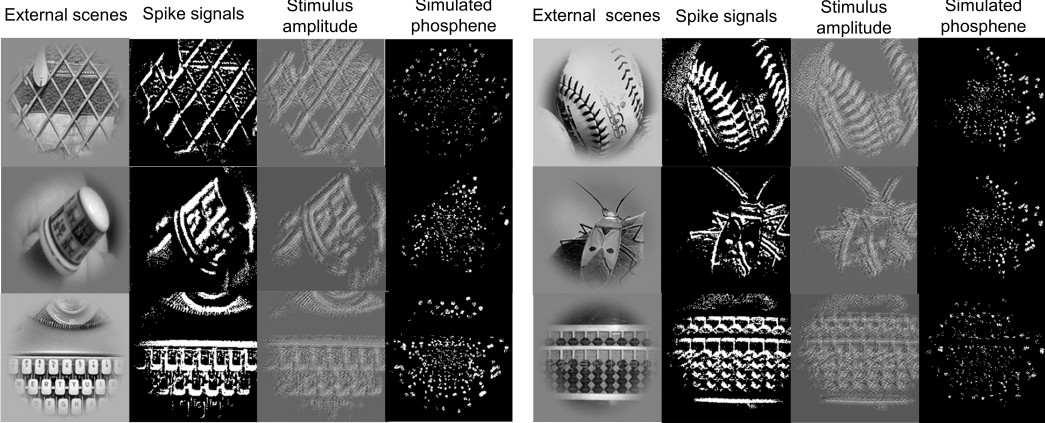

Figure 4: We randomly selected six external scenes from the test dataset to display the signal-processing process of StimuSEE. The external scene is converted to spike trains by a DVS sensor, and then the spike train within one-time window is constructed to a binary image to ensure its quality. The constructed binary image demonstrates the spike signals contain fruitful features, including edge and texture information. The predicted stimulation amplitude by the SRNN model also contains scene features, whose value contributes to a similar firing rate to that of normal neurons. The simulated phosphenes by a biological phosphene model will be transferred to the visual cortex for perception.

70% of the spike trains from the DVS sensor and corresponding recorded V1 neural responses were used for training. Another 10% was used to validate the model during training, and the remaining 20% were used to test the final performance.

**Data flow of StimuSEE.** The external scenes are recorded to sparse spike trains and then reconstructed into binary images to validate their quality. As is shown in Figure 4, the recorded spike signals contain fruitful features, including edge and texture information. Then, the predicted stimulus amplitude of the SRNN model has a similar shape with a constructed binary image of spike signals but has quite different intensity. During the animal experiment, the stimulus amplitude is an important factor affecting neurons' firing rate. In this framework, the stimulus amplitude is trained with supervised signals from the neural response of V1 cells, which makes the stimulation patterns more effective for vision restoration. The simulated phosphenes are obtained by the proposed biological simulation model, which is the input signal for PVS-net to fit the neural response of V1 neurons.

**Loss function.** In this study, we adopt the Mean absolute error (MAE) loss function to minimize the difference between the predicted firing rate of V1 neurons against the ground truth. We also applied an $L_2$ penalty to the Laplacian of the weights to encourage spatial locality and impose regular changes in space.

$$\mathcal{L}_{Smoothness} = \sqrt{\sum_i (W_i * L)}, L = \begin{bmatrix} 0 & -1 & 0 \\ -1 & 4 & -1 \\ 0 & -1 & 0 \end{bmatrix}. \qquad (10)$$

Thus, the total loss is summarized as follows:

$$\mathcal{L}_{total} = \mathcal{L}_{MAE} + \mathcal{L}_{Smoothness} \qquad (11)$$

**Training details.** The phosphene model is a biological model whose parameter is adjusted based on experimental data and kept fixed during training. The PVS-net is pre-trained with the CIFAR-10 dataset. The parameters of the PVS-net and the feature bank are updated with stopgradient operation. The StimuSEE is trained to fit the responses of the 166 neurons with the spike trains from the DVS sensor as input. The framework is implemented with Pytorch on NVIDIA RTX 3060. We use Adam optimizer with a base learning rate of $1e-3$, $\beta_1, \beta_2 = 0.9, 0.999$, and a weight decay of 0.01 to minimize the joint loss.

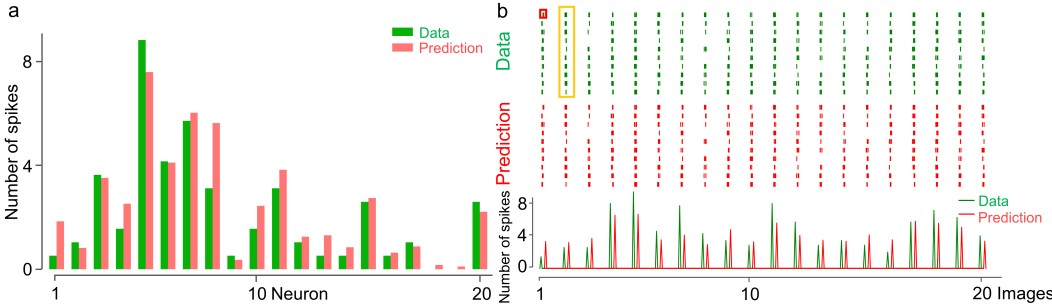

Figure 5: The performance of StimuSEE to fit the response of V1 neurons in the test set. (a) In the spatial domain, 20 neurons were randomly selected from 166 neurons to show their performance. The data represented the recorded firing rate of several V1 neurons when displaying the image to a monkey. The prediction in this figure means the predicted spike number of the StimuSEE framework. The predicted spike number of V1 neurons in one scene can fit well with the recorded response of normal cells. (b) In the temporal domain, the front 20 image prediction of StimuSEE in one selected neuron is compared with that of a normal cell. The spike train in the red rectangle was generated according to the corresponding firing rate of one neuron. This spike train's time distribution and spike number followed the Poisson distribution. In addition, this generated process is repeated ten times to avoid randomness, as shown in the yellow rectangle. The spike train of the StimuSEE framework adopts the same generated method. The comparison between the firing rate of recorded data and predicted values was depicted in the bottom column. The comparison in spike train and spike number proved the predicted results of StimuSEE could fit well with the recorded data in the temporal domain.

## 4.2 PERFORMANCE EVALUATION OF STIMUSEE

**Fitting the response of primary visual cortex.** The performance of stimulation patterns and simulated phosphenes are verified by fitting the spike response of 166 V1 neurons. We first measure the performance by calculating the difference between predicted values and recorded spike number of 166 neurons in all test images. The percentage of difference between [-0.5, 0.5] achieves 0.54 that represents a good fitting performance in all test dataset. Then, we measure the performance from both the spatial and temporal domains. The performance in the spatial domain is measured by the similarity between the predicted firing rate of 166 neurons to one scene and the recorded values of V1 cells. The similarity between two values is measured by Pearson correlation coefficients (PCC). The percentage of PCC in all test scenes that is above 0.75 exceeds 80%. In addition, we select 20 neurons' responses to one scene to demonstrate its performance, where it has high similarity between predicted value and recorded response, including cells with high and low spike firing rates (Figure 5(a)).

The temporal domain performance is measured by the similarity between the single neuron's response to various external scenes and corresponding predicted values. The proportion of Pearson correlation coefficients (PCC) between these two values that exceeds 0.75 is approximately 0.69. To better illustrate its similarity, the generated spike trains for 20 scene images are depicted in the first column, as shown in Figure 5(b). These spike trains were generated based on the recorded firing rate of a randomly selected V1 neuron. The spike train in the red rectangle displays the spike number and time distribution in 660 ms, corresponding to 20 time steps. This generated process follows Poisson distribution, repeated ten times to avoid randomness. The spike train of the SRNN model adopts the same method. The comparison results show that the spike train of the SRNN model can fit well with the recorded response. Then, the bottom column compares the spikes between those two values. The comparison results prove that the predicted values can fit well with the number of spikes of single cell for various scenes. In addition, the performance of fitting results achieves 0.78 in the test dataset evaluated by the Pearson correlation coefficient.

**Comparison with State-of-the-art works.** The proposed StimuSEE verifies the quality of stimulation patterns by comparing the predicted firing rate of V1 neurons with that of normal neurons. Other existing works either neglect the distortion effects, which are modeled by Phosphene model,

Table 1: Comparison of performance between proposed StimuSEE and other prediction models

| Reference | (Yan et al., 2020) | (Cadena et al., 2019) | (Wang et al., 2022b) | **This work** |
|---|---|---|---|---|
| Processing model | CNN | CNN | SNN | **SRNN** |
| Validate component | N/A | N/A | N/A | **BSM & PVS-net** |
| Target cells | Ganglion cells | V1 cells | V1 cells | **V1 cells** |
| PCC | 0.7 | 0.72 | 0.65 | **0.78** |

BSM = Biological simulation model;

Table 2: Comparison of power consumption between SRNN and CNN model with similar architecture

| Model | First layer | | Second layer | | Recurrent layer | Total | |
|---|---|---|---|---|---|---|---|
| | Firing rate | Power | Firing rate | Power | Power | SFR | Power |
| SRNN | 4.61% | 11 $\mu J$ | 9.97% | 11.65 $\mu J$ | 0.13 $\mu J$ | 7.29% | 22.78 $\mu J$ |
| CNN | N/A | 59 $\mu J$ | N/A | 59.6 $\mu J$ | N/A | N/A | 118.6 $\mu J$ |

or verify stimulation qualities by image reconstruction quality. We argue that these approaches lack feasibility for real-world retinal prostheses. Moreover, StimuSEE predicts the stimulation patterns with the SRNN model, whose spike-based processing characteristic makes it energy-efficient and more suitable for retinal prosthesis. See Table A2 for a detailed comparison of StimuSEE and existing works.

Although the primary contribution of this work is not to design a framework that boosts the performance of predicting V1 cells firing rate, we still compare with some works that only focus on predicting the neural response of ganglion cells or V1 cells for reference (Table 1). These frameworks achieve 0.65-0.72 with the evaluation method by Pearson correlation coefficient. However, in our framework, we utilize the SRNN model to obtain stimulation patterns, the phosphene model to simulate phosphene in the retina, PVS-net to fit the neural response. It still achieves 0.78 in the final fitting tasks, which proves that this framework can obtain more effective stimulation patterns and has a high potential to improve the restoration effect for blind people.

**Power Consumption Analysis of SRNN model.** We evaluate the energy efficiency of the proposed SRNN model because only this model needs to be deployed into an implanted prosthesis for prediction. As shown in Table 2, the spike firing rates (SFR) of the first and second spike layers are 4.61 % and 9.97 %, respectively. The overall SFR is 7.29 %. This low firing rate characteristic can significantly decrease the amount of calculation and power consumption. We compare the power consumption of the SRNN model with a CNN model of similar architecture where the power consumption of multiplication and addition with 32-bit is 3.7 pJ and 0.9 pJ, respectively (Horowitz, 2014). From Table 2, we see that for the SRNN model, the energy of the first, second, and recurrent layers are 50.1 $\mu J$, 749 $\mu J$, and 51.9 $\mu J$. The total energy of the SRNN model in one prediction is 851 $\mu J$, around five times lower than the CNN model, which consumes 3.982 $mJ$ in one prediction.

## 5 CONCLUSION

In this study, we propose for the first time an end-to-end stimulation patterns optimization framework (StimuSEE) for retinal prostheses. StimuSEE consists of a retina network, a phosphene model, and a PVS-net to predict stimulation patterns and validate its effectiveness. Besides, it adopts V1 neurons' spike response as supervision during training, which provides a better optimization direction to obtain effective stimulation patterns for vision restoration. The predicted stimulation patterns can generate a firing rate similar to V1 cells, whose performance evaluated by Pearson correlation coefficient achieves 0.78. In sum, the proposed StimuSEE is a promising processing framework to obtain effective stimulation patterns to empower visual prostheses for better vision restoration.

## 6 ACKNOWLEDGMENTS

This work was supported by the STI2030-Major Project (Grant No. 2022ZD0208805) and Key Project of Westlake Institute for Optoelectronics (Grant No. 2023GD004).

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
