## APPENDIX

### A. VISION REHABILITATION MECHANISM OF RETINAL PROSTHESES

EPIRET3 is adopted to introduce the working principle of retinal prostheses Klauke et al. (2011); Vanhoestenberghe & Donaldson (2013). The prosthesis consists of a receiver coil, receiver chip, stimulation chip, and stimulation electrodes, as shown in Figure A1(a). The stimulation pattern from external processing unit is a simple shape, like a line and circle, which transmits to the prosthesis through transmitter and receiver coils. Then, the receiver chip and stimulation chip process these received signals and provide the control signals for the electrode array. The electrode array is composed of 25 electrodes. It is necessary to explore the suitable pulse duration and current amplitude of the electrode array before stimulation (Figure A1(b)). The experiment results demonstrated that the stimulation pulse amplitude has a strong relationship with visual perception Sekhar et al. (2017). Then, the restoration effect of this device is verified in several RP patients. The visual perception of these patients after stimulation is recorded in Figure A1(c). Although the prosthesis still has the problem of inconsistent stimulation effect and poor repetition effect, the patient does produce visual perception via this device.

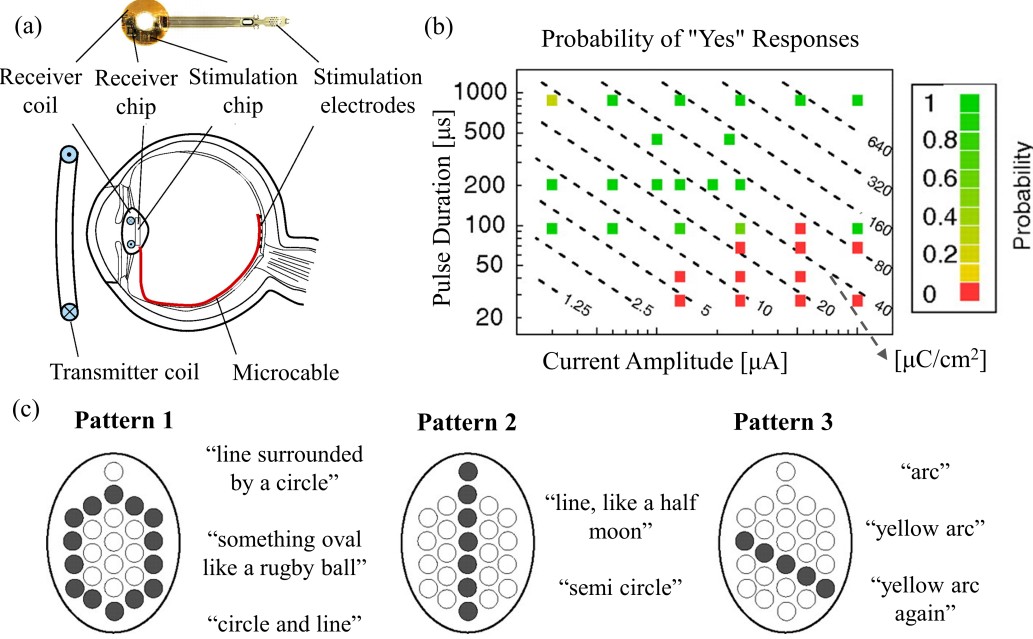

Figure A1: The overall architecture and technique details of EPIRET3. (a) The implant includes a receiver coil, receiver chip, stimulation chip, and stimulation electrodes. (b) The pulse duration and current amplitude are explored to the probability of neural response. Red color means no response, while blue color represents the high response. (c) Three different stimulation patterns are used to stimulate ganglion cells of several RP patients. The right part is the visual perception of the patient after stimulation. (Figure reproduced from ref Klauke et al. (2011))

### B. SPIKE REPRESENTATION ENCODING TECHNIQUE

The spike representation encoding technique is proposed to convert external scenes into sparse spike trains. This process is usually composed of spike signal recording and signal processing. A synchronous recording system displays external information and collect corresponding spike signals. Figure A2 (a) shows that the recording system includes display and record modules. When the recording begins, the display part displays the scene picture. To maximize the preservation of detail features, the scene picture is moved in the direction of 45 angles to the X and Y axis (Figure A2 (b)). The recording part utilizes a dynamic vision sensor to record the feature of the scene with spike train. The display part would suspend for 1000 ms for the record part to store the spike train. Then,

this system moves to display and records the next scene picture. The synchronization between two modules in this system is ensured utilizing TCP/IP protocol.

The signal processing is to prepare the input spike events for the SRNN model with the recorded spike train. First, the recorded spike trains are mixed with environmental noise that should be removed. A denoising method proposed by Serrano-Gotarredona was adopted to remove the noise and improve the quality of the spike train (Leñero-Bardallo et al. (2011)). Then, this long spike train is divided into several spike trains, making it suitable as the input data for the SRNN model. In this experiment, the time window of each spike train is set to 33 ms, which can provide effective scene features for processing. In addition, the spatial resolution of the spike train is decreased to 200 × 200 pixels for the SRNN model.

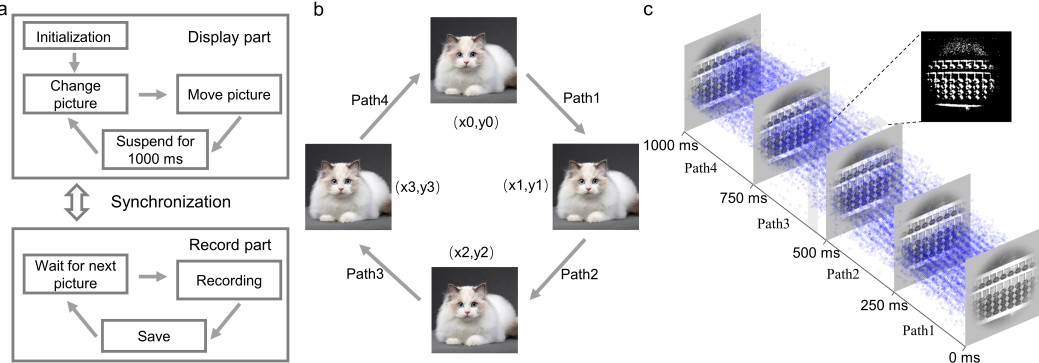

Figure A2: The details about spike representation encoding technique. (a) The synchronous recording system, including the record and display modules, is designed to record the spike train of external scenes. (b) The scene picture is moving in the direction of 45 angles to the X and Y axis. (c) The spike train of one-time window is constructed to a binary image that contains scene information, like edge and texture features.

The recorded spike train of one scene picture is depicted in Figure A2(c). To verify the quality of the spike train, a binary image is built with the spike train in one-time window. This image has clear edge and shape information, providing effective input signals for the SRNN model.

## C. THE PERFORMANCE OF STIMUSEE

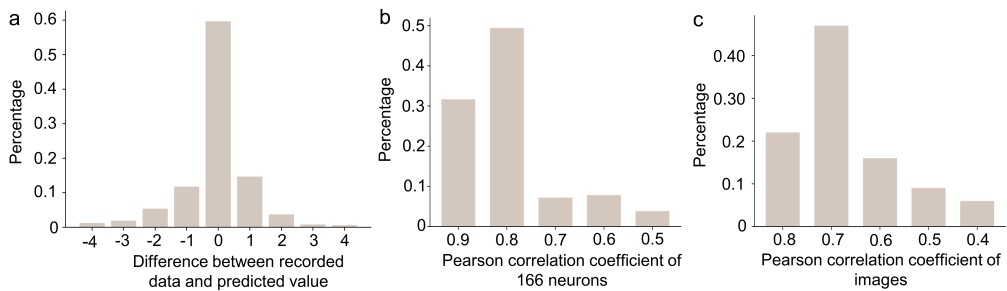

Figure A3: The performance of StimuSEE to fit the response of V1 neurons in the test set. (a) The distribution of difference between predicted values and recorded spike number of 166 neurons in all test images. The difference between [-0.5, 0.5], [-1.5, -0.5], and [0.5, 1.5] are 0.60, 0.13, and 0.10, respectively. (b) In the spatial domain, the distribution of Pearson correlation coefficient of 166 neurons to each image. (c) In the temporal domain, the distribution of Pearson correlation coefficient of each neuron to all test images.

The StimuSEE can predict the response of V1 neurons, whose performance can be demonstrated by the difference between the predicted values and recorded neurons' response (Figure A3(a)). The difference between [-0.5, 0.5], [-1.5, -0.5], and [0.5, 1.5] are 0.60, 0.13, and 0.10, respectively. Its

results prove that StimuSEE can fit well with the response of V1 neurons. The performance of StimuSEE is also evaluated in spatial and temporal domains. As shown in Figure A3(b), the values of 0.9 and 0.8 on the Y-axis represent Pearson correlation coefficients that exceed 0.85 and fall within the range of [0.75, 0.85]. Additionally, the proportion of values within this range reaches 0.8. As shown in Figure A3(c), the values of 0.8 and 0.7 on the Y-axis represent Pearson correlation coefficients that exceed 0.75 and fall within the range of [0.65, 0.75], and the proportion of values within this range reaches 0.69. These results prove the predicted values has higher similarity in the spatial and temporal domains.

## D. PERFORMANCE WITH DIFFERENT LOSS FUNCTION

Table A1: The performance of StimuSEE with different loss function on test dataset

|  | Only loss function | Loss with smoothness | Loss with L1 regularization | Joint loss |
|---|---|---|---|---|
| MAE loss function | 0.72±0.21 | 0.72±0.18 | 0.76±0.12 | 0.78±0.09 |
| MSE loss function | 0.71±0.25 | 0.72±0.20 | 0.74±0.16 | 0.77±0.06 |
| Poisson loss function | 0.69±0.30 | 0.69±0.23 | 0.70±0.16 | 0.73±0.12 |

The performance of StimuSEE to fit neurons' response is variable towards different loss functions. As shown in Table A1, the performance increases as the framework adopts weight smoothness and L1 regularization in the training process, which is effective to solve overfitting problem. In addition, the performance of the Mean Absolute Error (MAE) loss function is similar to the Mean Squared Error (MSE), better than the Poisson loss function. The Poisson loss function is suitable for targets that follow Poisson distribution. This experiment's targets do not follow Poisson distribution but are highly related to the external scene stimulus. In sum, the StimuSEE with joint loss function achieved 0.78 in fitting the multiple neurons' response Yan et al. (2020); Cadena et al. (2019).

## E. THE LIMITATION OF THIS RESEARCH

This study presents a suitable processing framework to obtain effective stimulation patterns for retinal prostheses, which can still be improved in several aspects. The phosphene model can be improved by collecting more patient data for better simulation. StimuSEE can also benefit from switching from adopting static scenes as external signals to dynamic videos (Wang et al., 2022a). Finally, the firing rate of 166 V1 neurons is adopted as the label to train the processing framework. It would also be meaningful to utilize the responses of more V1 neurons as labels in the training process. Apart from improving the performance of the processing framework, it is necessary to deploy it in prostheses and design an animal experiment to verify the effectiveness of the stimulation patterns (Bashivan et al., 2019; Beauchamp et al., 2020). In the animal experiment, the recorded responses of V1 neurons in a blind animal can be compared with those of a sighted animal (Mathieson et al., 2012). Then, the processing framework can adjust the stimulation patterns based on the observed differences for better fitting performance (Ponce et al., 2019).

Table A2: Comparison of main features between proposed StimuSEE and existing works

| Reference | (Guo et al., 2018) | (Burcu et al., 2022) | (Granley et al., 2022) | **This work** |
|---|---|---|---|---|
| Processing model | Saliency detection | CNN | CNN | **SRNN** |
| Phosphene model | N/A | SVM | BSM | **BSM** |
| Verification method | N/A | Image | Image | **Neuron response** |

SVM = Stimulation vector mapping; BSM = Biological simulation model;