# OpenReview forum: "Exploring Effective Stimulus Encoding via Vision System Modeling for Visual Prostheses"
_ICLR.cc/2024/Conference — ICLR 2024 poster_

### Official Review · Reviewer_JqK3 · 2023-10-28

**Soundness:** 3 good
**Presentation:** 2 fair
**Contribution:** 4 excellent
**Rating:** 6
**Confidence:** 3

**Summary:**

This research proposes a new end-to-end architecture that can be used to generate an appropriate electrical stimulation pattern of visual prostheses based on an input natural image through a network of SRNNs, which can then be passed through a network of CNNs to induce the desired V1 neuronal activity. If this information flow can be successfully implemented, the network can be used to generate a corresponding electrical stimulation pattern based on the image. This would have great applications in patients with retinal dysfunction.

Overall, the idea is great and I see great potential for this approahc in developing useful visual prostheses.

**Strengths:**

1. End-to-end prediction of V1responses using visual prothesis is rare.
2. Uses an SRNN model with low energy cost is used.
3. The design of the overall framework is elegant.

**Weaknesses:**

The overall writing is good, but some technical details are not clear

**Questions:**

1. What is the adjusted ReLU function for surrogate gradient?
2. What are the differences between LIF and MP_LIF ?
3. How are the parameters of the phosphene model selected? It says "fitting the experimetnal data recorded from patients with retinal protheses". I didn't see any fitting procedure or decription of the data.
4. Table 2, The performance of this work is better than CNN approach (Cadena et al, 2009). But can you show the number of parameters of each approach?
5. Table 3 shows the energy cost of SRNN and CNN with a similar achitechture. The advange of SRNN is not surprising given the intrisically low energy cost of SRNN. However, the energy cost of SRNN should be compared to that of a CNN with similar performance (such as, Cadena, 2009) rather than similar achitechture.

---

> ### Author Response · Authors · 2023-11-20
> **Response to Reviewer JqK3**
>
> **Q1: What is the adjusted ReLU function for surrogate gradient?**
>
> **A1:** The adjusted ReLU function is defined as:
>
> $$f(x) = ln(1+e^{(x-\eta)}),$$ where $\eta$ is the threshold to control the intensity in EQ. (4) of the original manuscript.
>
> **Q2: What are the differences between LIF and MP_LIF ?**
>
> **A2:** The output of the LIF neuron is a spike train, while the output of the MP_LIF neuron is the numerical values of membrane potentials. Naively speaking, the LIF neurons output 'where the spikes are' and  MP_LIF neurons output 'how large is the membrane potentials'. The predicted membrane potentials will be set to zero after a spike is fired.
>
> **Q3: How are the parameters of the phosphene model selected? It says "fitting the experimetnal data recorded from patients with retinal protheses". I didn't see any fitting procedure or decription of the data.**
>
> **A3:** The hyperparameters of the phosphene model utilized in this work are based on various experimental and modeling results from different components of corresponding research:
>
> In Eq. 2, we use $a = 0.75$, $b = 120$, $k = 1.73$, based on a fitting method of [1] on data of a human experiment from [2].
>
> In Eq. 3, Eq. 4, $Pw = 3.3ms$, $f = 30Hz$, $threshold = 0.3 mW/mm^{2}$ are obtained from the setting and human experimental results with retinal prosthesis [3]. We use $I_{leak} = 23.9 \mu A$ based on a data fitting from [4].
>
> In Eq. 5, Eq. 6, $a_{1}$, $a_{2}$, $b_{1}$, and $b_{2}$, $\lambda$ is adopted from a retinal phosphene model [5].
>
> In Eq. 7, $d_{s}$, $d_{e}$, $\rho$, and $\gamma$ were obtained from a human retina model [6].
>
> **Q4: Table 2, The performance of this work is better than CNN approach (Cadena et al, 2019). But can you show the number of parameters of each approach?**
>
> **A4:** The number of total parameters in our proposed StimuSEE, including retina-net and PVS-net is 68.9 K, while the number of parameters in the compared CNN approach is 62 K.
>
> **Q5: Table 3 shows the energy cost of SRNN and CNN with a similar achitechture. The advange of SRNN is not surprising given the intrisically low energy cost of SRNN. However, the energy cost of SRNN should be compared to that of a CNN with similar performance (such as, Cadena, 2019) rather than similar achitechture.**
>
> **A5:** In the proposed StimuSEE, we predict stimulation patterns using **spike trains** recorded from a dynamic vision sensor as input. The frameworks designed by Cadena et al. [7] and Yan et al. [8] take images as input and output responses of V1 cells and Ganglion cells, respectively. Therefore, directly comparing power consumption under the same performance is intractable given different inputs and outputs. In Tab. 3, we demonstrate that our designed StimuSEE is energy efficient, leading to a more practical visual prosthesis.
>
> # References
>
> [1] Polimeni J R, Balasubramanian M, Schwartz E L. Multi-area visuotopic map complexes in macaque striate and extra-striate cortex[J]. Vision research, 2006, 46(20): 3336-3359.
>
> [2] Horton J C, Hoyt W F. The representation of the visual field in human striate cortex: a revision of the classic Holmes map[J]. Archives of ophthalmology, 1991, 109(6): 816-824.
>
> [3] Granley J, Beyeler M. A computational model of phosphene appearance for epiretinal prostheses[C]//2021 43rd Annual International Conference of the IEEE Engineering in Medicine & Biology Society (EMBC). IEEE, 2021: 4477-4481.
>
> [4] Fernández E, Alfaro A, Soto-Sánchez C, et al. Visual percepts evoked with an intracortical 96-channel microelectrode array inserted in human occipital cortex[J]. The Journal of clinical investigation, 2021, 131(23).
>
> [5] Granley J, Relic L, Beyeler M. Hybrid neural autoencoders for stimulus encoding in visual and other sensory neuroprostheses[J]. Advances in Neural Information Processing Systems, 2022, 35: 22671-22685.
>
> [6] Jansonius N M, Nevalainen J, Selig B, et al. A mathematical description of nerve fiber bundle trajectories and their variability in the human retina[J]. Vision research, 2009, 49(17): 2157-2163.
>
> [7] Cadena, Santiago A., et al. "Deep convolutional models improve predictions of macaque V1 responses to natural images." PLoS computational biology 15.4 (2019): e1006897.
>
> [8] Yan, Qi, et al. "Revealing fine structures of the retinal receptive field by deep-learning networks." IEEE transactions on cybernetics 52.1 (2020): 39-50.

---

### Official Review · Reviewer_NiJW · 2023-10-31

**Soundness:** 3 good
**Presentation:** 3 good
**Contribution:** 3 good
**Rating:** 6
**Confidence:** 3

**Summary:**

This paper introduces an end-to-end framework (StimuSEE) designed for visual prostheses through the integration of a retinal network, a phosphene model, and a primary visual system network (PVS-net). Specifically, this method generates stimulation patterns by using the corresponding V1 neuron spike patterns as supervision. The performance of this framework is impressive (Pearson correlation coefficient = 0.78 between the predicted and groundtruth firing rate).

**Strengths:**

Overall, I think this paper showcases a notable contribution to the field, especially with its V1 neural encoding approach for visual prostheses. The writing is clear and well-organized. The methods and results are solid. The work has novelty and can lead to potential clinical applications, such as restoring vision for the blind.

**Weaknesses:**

Some details in methods and results need to be further elaborated in the main text.
I am willing to raise the score if the authors could provide clarification to my questions listed below.

**Questions:**

**Questions**:
- It's unclear to me what is the novelty of the proposed phosphene model compared to Granley et al. (2022).
- "The training details" section requires further elaboration to enhance clarity for readers and to ensure the work is reproducible by other researchers, for example:
    - How is PVS-net pre-trained? Was the same dataset utilized both for the PVS-net pretraining and the end-to-end training?
    - What is the temporal resolution of the label (neural firing rate)? How many time points per image?
- Since the key innovation of this paper is using the V1 neural prediction instead of image reconstruction as the objective, a more detailed discussion and presentation of results in Fig.5 would be beneficial to convey the claims:
    - In Fig 5a, the authors only showcased 20 neurons. It would be informative if the authors could provide quantitive results on the model's fit across all 166 neurons. Is Fig. 5b showing the results from a single neuron? If so, is it the best-performing one?
    - What would be the chance-level performance for both Fig. 5a and Fig. 5b?
    - Is Fig. 5a showing spike counts from 20 neurons for a single time point or single image? If yes, why not show the overall statistics for all images in the test dataset?
    - Fig. 5b shows the neural spike train over time. What is the temporal resolution here? Also, why comparing the spike train randomly generated 10 times from the firing rate, instead of directly comparing the time series of the predicted and groundtruth firing rate?
- How did you get the PCC in Table 2? Is it the correlation between the predicted and groundtruth firing rate along time? then averaged across all 166 neurons?

Minor point:
- Some fonts in equations (8) and (9) are not standard. Typically, functions or operators such as min, max, and ReLU are not italicized.
- It will be more reader-friendly to explain some abbreviations (e.g., DVS, LIF) in the caption of Fig. 2.

---

> ### Author Response · Authors · 2023-11-20
> **Response to Reviewer NiJW (1/2)**
>
> **Q1: It's unclear to me what is the novelty of the proposed phosphene model compared to Granley et al. (2022).**
>
> **A1:** Compared to the phosphene model in Granley et al. [1], the phosphene model of StimuSEE utilizes coordinate system transition for a more accurate phosphene location calculation. Also, the leakage $I_{leak}$ during the electrical stimulation process is also taken into account in the calculation of stimulation intensity. Furthermore, our proposed phosphene model incorporates a threshold function to determine the generation of phosphene.
>
> **Q2: How is PVS-net pre-trained? Was the same dataset utilized both for the PVS-net pretraining and the end-to-end training?**
>
> **A2:** We utilized CIFAR-10 dataset to pretrain the PVS-net with a classification task, and adopted the DVS-V1 dataset in the end-to-end training process.
>
> **Q3: What is the temporal resolution of the label (neural firing rate)? How many time points per image?**
>
> **A3:** The visual stimulus is presented to the subject to capture neural responses within a 60 ms time window. The spike firing rate is assessed by counting neuron firings within a fixed recording window of 60 ms. For one image, as the recording covers 166 neurons, the label consists of 166 numbers, each representing the corresponding neuron's firing count.
>
> **Q4: In Fig 5a, the authors only showcased 20 neurons. It would be informative if the authors could provide quantitive results on the model's fit across all 166 neurons. Is Fig. 5b showing the results from a single neuron? If so, is it the best-performing one?**
>
> **A4:** We first present the image-level Pearson Correlation Coefficient (PCC) distribution for all 166 neurons in Table 1. Specifically, for each image, we compute the PCC of the firing rate across all 166 neurons with respect to the ground truth, providing statistical information for all testing images. Additionally, in Table 2, we offer the neuron-level PCC distribution for all 166 neurons. This involves calculating the PCC of the firing rate for each individual neuron in relation to the ground truth across all testing images, and presenting statistics for all 166 neurons. These experimental results show that our proposed StimuSEE fits the neural response of V1 cells well. Fig. 5b shows one of the best-performing ones.
>
> Table 1: The distribution of (PCC) of 166 neurons to each image
> | PCC | >0.85 | 0.75-0.85 | 0.65-0.75 | 0.55-0.65 | <0.55 |
> | :----: | :----: | :----: | :----: | :----: | :----: |
> | Percentage | 0.32| 0.50 | 0.08 | 0.07 |  0.03 |
>
> Table 2:  The distribution of PCC of each neuron against all test images
> | PCC | >0.75 | 0.65-0.75 | 0.55-0.65 | 0.45-0.55 | <0.45 |
> | :----: | :----: | :----: | :----: | :----: | :----: |
> | Percentage | 0.22| 0.47 | 0.16 | 0.09 |  0.06 |
>
> **Q5: What would be the chance-level performance for both Fig. 5a and Fig. 5b?**
>
> **A5:** We provide statistics Tab. 3 and 4 as in **A4** based on our defined chance-level predictor.
>
> Table 3: The distribution of PCC of 166 neurons to each image
> | PCC | >0.15 | 0.05 - 0.15 | -0.05 - 0.05 | -0.15 - -0.05 | <-0.15 |
> | :----: | :----: | :----: | :----: | :----: | :----: |
> | Percentage | 0.04| 0.25 | 0.48 | 0.21 |  0.02 |
>
> Table 4:  The distribution of PCC of each neuron against all test images
> | PCC | >0.15 | 0.05 - 0.15 | -0.05 - 0.05 | -0.15 - -0.05 | <-0.15 |
> | :----: | :----: | :----: | :----: | :----: | :----: |
> | Percentage | 0.07| 0.24 | 0.42 | 0.22 |  0.05 |
>
> **Q6: Is Fig. 5a showing spike counts from 20 neurons for a single time point or single image? If yes, why not show the overall statistics for all images in the test dataset?**
>
> **A6:** Please refer to **A4** for the overall statistics.
>
> **Q7: Fig. 5b shows the neural spike train over time. What is the temporal resolution here? Also, why comparing the spike train randomly generated 10 times from the firing rate, instead of directly comparing the time series of the predicted and groundtruth firing rate?**
>
> **A7:**  We apologize for any confusion caused by the x-axis of Figure 5b. When referring to 'times,' we meant images. Figure 5b illustrates the firing pattern of a single neuron across the first 20 test images (x-axis). The temporal resolution of both the labeled and predicted neuron responses is modeled using a Poisson distribution. The fitting process is iterated ten times to mitigate the impact of randomness. We will revise Fig. 5b in the camera-ready version for better presentation.
>
> **Q8: How did you get the PCC in Table 2? Is it the correlation between the predicted and ground truth firing rate along time? then averaged across all 166 neurons?**
>
> **A8:** Since the spike firing rate is assessed by counting neuron firings within a fixed recording window of 60 ms. The label consists of 166 numbers, each representing the corresponding neuron's firing count. Thus, we can directly calculate the PCC of all 166 neurons.

---

> > ### Author Response · Authors · 2023-11-20
> > **Response to Reviewer NiJW (2/2)**
> >
> > **Q9: Some fonts in equations (8) and (9) are not standard. Typically, functions or operators such as min, max, and ReLU are not italicized.**
> >
> > **A9:** The font errors are fixed in the camera-ready version.
> >
> > **Q10: It will be more reader-friendly to explain some abbreviations (e.g., DVS, LIF) in the caption of Fig. 2.**
> >
> > **A10:** We have included the full forms of these abbreviations in the caption of Fig. 2.
> >
> > # References
> >
> > [1] Granley, Jacob, Lucas Relic, and Michael Beyeler. "Hybrid neural autoencoders for stimulus encoding in visual and other sensory neuroprostheses." Advances in Neural Information Processing Systems 35 (2022): 22671-22685.

---

> > > ### Comment · Reviewer_NiJW · 2023-11-22
> > >
> > > Thanks for your reply and clarification. I don't have further feedback.

---

### Official Review · Reviewer_Fc3Q · 2023-11-01

**Soundness:** 4 excellent
**Presentation:** 3 good
**Contribution:** 4 excellent
**Rating:** 8
**Confidence:** 4

**Summary:**

This paper proposes an end-to-end framework called StimuSEE for visual prosthetics. The proposed frameworks makes use of v1 neuron responses as feedback signal to train a retinal network to generate stimulation pattern. Their results show that with this framework, they could general meaningful simulation patterns that could predict v1 neurons with substantial performance.

**Strengths:**

The approach proposed in the paper is novel to my knowledge. The training pipeline is simplistic and clever.
The paper itself is well written, and thorough about providing experimental details for future replication.

**Weaknesses:**

Though the framework demonstrate reliable results in predicting V1 neurons, because of the different performance metric (neuronal responses vs. reconstruction error), it is hard to directly compare this method against other existing methods. One potential solutions is adding a decoder layer on top of the trained StimuSEE framework and see whether it indeed also facilitate reconstruction as well.

Since the StimuSEE frameworks consists of multiple models, the paper perhaps should make it more clear which of the models are trained with the error signals from predicting V1 responses. For example, in Figure 2, it could be helpful to use a bounding box with a different color or the SRNN model or an arrow signifying the direction of training feedback.

**Questions:**

The paper only briefly mentions that V1 neuron response in blind animals are comparable to those in sighted animals, it is unclear how this approach directly transfers to blind animals. Does this framework end up with different learned model based on coverage of V1 neurons in recording? Could you use two sighted animal to estimate how transferrable the model is from one animal to the other? Also since the frameworks requires many pre-determined parameters for other non-trainable models, could the authors provide some insight on how sensitive is the SRNN network training with respect to different parameter setting among the other models?

Could you elaborate on the purpose of the parameter index in equations (8)?

---

> ### Author Response · Authors · 2023-11-20
> **Response to Reviewer Fc3Q (1/2)**
>
> **Q1: Though the framework demonstrates reliable results in predicting V1 neurons, because of the different performance metrics (neuronal responses vs. reconstruction error), it is hard to directly compare this method against other existing methods. One potential solution is adding a decoder layer on top of the trained StimuSEE framework and see whether it indeed also facilitates reconstruction as well.**
>
> **A1:** We could potentially replace StimuSEE's PVS-net with a decoder for image reconstruction. However, unlike other existing frameworks (hybrid neural autoencoders) that use the original image as input for reconstruction, our proposed StimuSEE utilizes spike trains recorded from a dynamic vision sensor as input. From an information perspective, reconstructing the original image with spike train input is a more challenging task compared to using the original image as input.
>
> **Q2: Since the StimuSEE frameworks consist of multiple models, the paper perhaps should make it more clear which of the models are trained with the error signals from predicting V1 responses. For example, in Figure 2, it could be helpful to use a bounding box with a different color or the SRNN model or an arrow signifying the direction of training feedback.**
>
> **A2:** The parameters of the retina-net and PVS-net are updated based on the error signals during model training, and the parameters of the phosphene model are kept fixed. As suggested, we will revise Fig. 2 to better indicate the gradient flow.
>
> **Q3:The paper only briefly mentions that V1 neuron responses in blind animals are comparable to those in sighted animals, it is unclear how this approach directly transfers to blind animals.**
>
> **A3:**  The concept of using visual cortical prostheses for vision restoration has long been proposed. Beauchamp et al. [1] demonstrated that dynamic stimulation could facilitate the recognition of letter shapes in both sighted and blind individuals. Building upon this insight, we introduce StimuSEE, a system that leverages the V1 neuron response in sighted animals as a reference to create stimulation for the blind. In parallel, researchers [2,3] have explored the correlation between stimulation and the responses of vision neurons.
>
> **Q4: Does this framework end up with different learned model based on coverage of V1 neurons in recording? Could you use two sighted animal to estimate how transferrable the model is from one animal to the other?**
>
> **A4:** We appreciate Reviewer Fc3Q for posing these intriguing open questions. Given that the dataset employed in this study is obtained from a single subject with a fixed electrode placement corresponding to one coverage pattern of V1 neurons, direct answers to these questions are not feasible. Considering that StimuSEE generates stimulation patterns based on V1 neuron firing, we posit that transitioning to a different V1 coverage pattern would result in a different model. The verification of transferability among animals will be addressed in our future research endeavors.
>
> **Q5: Since the frameworks require many pre-determined parameters for other non-trainable models, could the authors provide some insight on how sensitive is the SRNN network training with respect to different parameter settings among the other models?**
>
> **A5:**  The hyperparameters of the phosphene model utilized in this work are based on various experimental and modeling results from different components of corresponding research:
>
> In Eq. 2, we use $a = 0.75$, $b = 120$, $k = 1.73$, based on a fitting method of [4] on data of a human experiment from [5].
>
> In Eq. 3, Eq. 4, $Pw = 3.3ms$, $f = 30Hz$, $threshold = 0.3 mW/mm^{2}$ are obtained from the setting and human experimental results with retinal prosthesis [6]. We use $I_{leak} = 23.9 \mu A$ based on a data fitting from [7].
>
> In Eq. 5, Eq. 6, $a_{1}$, $a_{2}$, $b_{1}$, and $b_{2}$, $\lambda$ is adopted from a retinal phosphene model [8].
>
> In Eq. 7, $d_{s}$, $d_{e}$, $\rho$, and $\gamma$ were obtained from a human retina model [9].
>
> The phosphene model demonstrates robustness across various hyperparameter combinations, and the framework consistently produces similar prediction performance [10,11].
>
> **Q6: Could you elaborate on the purpose of the parameter index in equations (8)?**
>
> **A6:** Eq. 8 and Eq. 9 reduce the correlation among different neurons when predicting spike firing rates, simultaneously acting as a form of regularization.

---

> ### Author Response · Authors · 2023-11-20
> **Response to Reviewer Fc3Q (2/2)**
>
> # References
>
> [1] Beauchamp, Michael S., et al. "Dynamic stimulation of visual cortex produces form vision in sighted and blind humans." Cell 181.4 (2020): 774-783.
>
> [2] Walker, Edgar Y., et al. "Inception loops discover what excites neurons most using deep predictive models." Nature neuroscience 22.12 (2019): 2060-2065.
>
> [3] Bashivan, Pouya, Kohitij Kar, and James J. DiCarlo. "Neural population control via deep image synthesis." Science 364.6439 (2019): eaav9436.
>
> [4] Polimeni J R, Balasubramanian M, Schwartz E L. Multi-area visuotopic map complexes in macaque striate and extra-striate cortex[J]. Vision research, 2006, 46(20): 3336-3359.
>
> [5] Horton J C, Hoyt W F. The representation of the visual field in human striate cortex: a revision of the classic Holmes map[J]. Archives of ophthalmology, 1991, 109(6): 816-824.
>
> [6] Granley J, Beyeler M. A computational model of phosphene appearance for epiretinal prostheses[C]//2021 43rd Annual International Conference of the IEEE Engineering in Medicine & Biology Society (EMBC). IEEE, 2021: 4477-4481.
>
> [7] Fernández E, Alfaro A, Soto-Sánchez C, et al. Visual percepts evoked with an intracortical 96-channel microelectrode array inserted in human occipital cortex[J]. The Journal of clinical investigation, 2021, 131(23).
>
> [8] Granley J, Relic L, Beyeler M. Hybrid neural autoencoders for stimulus encoding in visual and other sensory neuroprostheses[J]. Advances in Neural Information Processing Systems, 2022, 35: 22671-22685.
>
> [9] Jansonius N M, Nevalainen J, Selig B, et al. A mathematical description of nerve fiber bundle trajectories and their variability in the human retina[J]. Vision research, 2009, 49(17): 2157-2163.
>
> [10] Granley J, Relic L, Beyeler M. Hybrid neural autoencoders for stimulus encoding in visual and other sensory neuroprostheses[J]. Advances in Neural Information Processing Systems, 2022, 35: 22671-22685.
>
> [11] Van der Grinten M L, de Ruyter van Steveninck J, Lozano A, et al. Biologically plausible phosphene simulation for the differentiable optimization of visual cortical prostheses[J]. bioRxiv, 2022: 2022.12. 23.521749.

---

### Official Review · Reviewer_QbGv · 2023-11-01

**Soundness:** 2 fair
**Presentation:** 2 fair
**Contribution:** 2 fair
**Rating:** 5
**Confidence:** 3

**Summary:**

The authors explored a better way of encoding the image stimuli, using spiking recurrent network, phosphene generation model, building in many details of early visual system and used a V1 encoding model. Then they optimize the electric stimuli generation parameters. constrained by V1 recording data. The model reached competitive results for modeling V1 data.

**Strengths:**

###

- The authors provided adequate background literature for evaluating this work.
- The authors built in numerous biological relevant details to make this stimulation system works, which is laudable.
- The primate V1 prediction results seems quite competitive.
- The energy efficiency is one highlight for this method!

**Weaknesses:**

### Weakness

- The method and experimental procedure is still a bit vague to me, i.e. I’m not sure how data and gradient flow through this system and how this system trains. Correct me if I’m wrong, during training you use the DVS → spiking RNN → stimulation model → phospene model → PVS-net V1 model and then fit all parameters with the primate V1 data? Some parameters of the phosphene model are constrained by human patient data, is that also transferred to fit primate V1 data?
- Eq. 8 and Eq. 9, the notations are somewhat confusing, are capital `Index` and `index` the same thing? What does $\cdot$ and $[]$ mean in the Eq. 9?.  I’m also not sure about the dimension of any of the variables… are they scalar or vector integer or what? Please clarify these in the paragraph following it.
- The biological data validation figure seems a bit weak.
    - Usually you’d like to plot the scatter plot between predicted spikes and actual spikes for all images and summarize the correlation for each cell. Figure 5 a) looks nice but lack the population summary.
    - If the data presented Figure 5 b is real biological spike data from times, it seems overly sparse. Can you clarify the image stimulation onset and offset time schedule of it? —— if the no spike period is just gray screen stimulation, then the temporal accuracy is just defined by the stimuli time course, which doesn’t mean much.
    - Further the temporal precision between the data and the prediction in Figure 5b seems **overly high.** With 10 repetitions, i.e. it can predict all the cases where spikes are missing? Maybe I misunderstood what data means in this case, is it the firing rate data for 166 V1 neurons, with some Poisson spike generator attached? How could the model know the differences between these different runs?
    - Is Figure 5 training set or test set?
    - Also not sure about the bottom panel in Figure 5 b), the x axes `Times` means different images?
- The work is technical solid, but the contribution and impact seems limited to the neuroengineering domain, not sure the technique will fit or benefit the neurips community.

**Questions:**

### Questions

- Not quite clear to me what “*the front 20 times prediction of*” mean in Figure 5 (b) caption. What are the x-axes “Times” in this plot? is it seconds?
- Given the four repeats in your V1 data, could you calculate the noise ceiling / self consistency of V1 response itself? then you can kind of compare your obtained pearson correlation to that “ceiling”.
- How do you compute the energy efficiency of Spiking RNN? also by counting the multiplication and addition in the simulated neuron firing process? —— if that’s the case won’t that depend on how do you discretize time?
- For more data to train the model, [1] seems to be quite relevant.

    [1] Chen, X., Wang, F., Fernandez, E., & Roelfsema, P. R. (2020). Shape perception via a high-channel-count neuroprosthesis in monkey visual cortex. *Science*, *370*(6521), 1191-1196.

---

> ### Author Response · Authors · 2023-11-20
> **Response to Reviewer QbGv (1/2)**
>
> **Q1: The method and experimental procedure is still a bit vague to me, i.e. I’m not sure how data and gradient flow through this system and how this system trains. Correct me if I’m wrong, during training you use the DVS → spiking RNN → stimulation model → phospene model → PVS-net V1 model and then fit all parameters with the primate V1 data? Some parameters of the phosphene model are constrained by human patient data, is that also transferred to fit primate V1 data?**
>
> **A1:** The StimuSEE is composed of a retinal network, a phosphene model, and a primary visual system network (PVS-net). The retinal network utilizes dynamic vision sensor and spiking RNN to perceive and process the scene signals. Then, a phosphene model is designed to simulate phosphene in the retina. Finally, PVS-net is adopted to predict the firing rate of V1 neurons with the simulated phosphene. The phosphene model is fixed during training. The parameter of retinal network and PVS-net are trained to fit the neural responses of V1 cells.
>
> **Q2: Eq. 8 and Eq. 9, the notations are somewhat confusing, are capital Index and index the same thing? What does $\cdot$ and $[]$ mean in the Eq. 9?. I’m also not sure about the dimension of any of the variables… are they scalar or vector integer or what? Please clarify these in the paragraph following it.**
>
> **A2:** $x \in R^{1 \times 166}$ is the extracted feature of the PVS-net. $params \in R^{166 \times D}$ refers to a learnable matrix, where $D$ is set to 50 in this work. The selected parameters are further dot producted with the input value $x$ to yield the final prediction results. $\cdot$ denotes dot production and $[]$ refers to indexing operation. Eq. 8 and Eq. 9 reduce the correlation among different neurons when predicting spike firing rates, simultaneously acting as a form of regularization.
>
> **Q3: Usually you’d like to plot the scatter plot between predicted spikes and actual spikes for all images and summarize the correlation for each cell. Figure 5 a) looks nice but lack the population summary.**
>
> **A3:** We first present the image-level Pearson Correlation Coefficient (PCC) distribution for all 166 neurons in Table 1. Specifically, for each image, we compute the PCC of the firing rate across all 166 neurons with respect to the ground truth, providing statistical information for all testing images. Additionally, in Table 2, we offer the neuron-level PCC distribution for all 166 neurons. This involves calculating the PCC of the firing rate for each individual neuron in relation to the ground truth across all testing images, and presenting statistics for all 166 neurons. These experimental results show that our proposed StimuSEE fits the neural response of V1 cells well. Fig. 5b shows one of the best-performing ones.
>
> Table 1: The distribution of PCC of 166 neurons to each image
> | PCC | >0.85 | 0.75-0.85 | 0.65-0.75 | 0.55-0.65 | <0.55 |
> | :----: | :----: | :----: | :----: | :----: | :----: |
> | Percentage | 0.32| 0.50 | 0.08 | 0.07 |  0.03 |
>
> Table 2: The distribution of PCC of each neuron against all test images
> | PCC | >0.75 | 0.65-0.75 | 0.55-0.65 | 0.45-0.55 | <0.45 |
> | :----: | :----: | :----: | :----: | :----: | :----: |
> | Percentage | 0.22| 0.47 | 0.16 | 0.09 |  0.06 |
>
> **Q4: If the data presented Figure 5b is real biological spike data from times, it seems overly sparse. Can you clarify the image stimulation onset and offset time schedule of it? —— if the no spike period is just gray screen stimulation, then the temporal accuracy is just defined by the stimuli time course, which doesn’t mean much.**
>
> **A4:** The visual stimulus is presented to the subject to capture neural responses within a 60 ms time window. The spike firing rate is assessed by counting neuron firings within a fixed recording window of 60 ms. For one image, as the recording covers 166 neurons, the label consists of 166 numbers, each representing the corresponding neuron's firing count. The offset time is 940 ms.
> Figure 5b illustrates the firing pattern of a single neuron across the first 20 test images (x-axis). The temporal spike train of both the labeled and predicted neuron responses is modeled using a Poisson distribution. The fitting process is iterated ten times to mitigate the impact of randomness.

---

> > ### Author Response · Authors · 2023-11-20
> > **Response to Reviewer QbGv (2/2)**
> >
> > **Q5: Further the temporal precision between the data and the prediction in Figure 5b seems overly high. With 10 repetitions, i.e. it can predict all the cases where spikes are missing? Maybe I misunderstood what data means in this case, is it the firing rate data for 166 V1 neurons, with some Poisson spike generator attached? How could the model know the differences between these different runs?**
> >
> > **A5:** The true label (data in Figure 5b) of spike firing rate is given by neuron firing counts during recorded time during vision stimulus. No recording is done during offset (no vision stimulus). The temporal resolution is fitted using a Poisson distribution, which is repeated ten times. Since the model is deterministic. The Poisson fitting is applied to the output of the PVS-net.
> >
> > **Q6: Is Figure 5 training set or test set?**
> >
> > **A6:** The displayed results in Figure 5 are test set results.
> >
> > **Q7: Also not sure about the bottom panel in Figure 5 b, the x axes Times means different images?**
> >
> > **A7:** In Figure 5b, the x axes can be times or images. In the neural response recording experiment, the duration of recording is 60 ms. The offset time between two stimuli is 940 ms. The whole period of one image is 1000 ms. In Figure 5b, we demonstrate the firing pattern of a single neuron across 20 seconds, which also means 20 test images. We will revise Fig. 5b in the camera-ready version for better presentation.
> >
> > **Q8: The work is technical solid, but the contribution and impact seems limited to the neuroengineering domain, not sure the technique will fit or benefit the neurips community.**
> >
> > **A8:** This study introduced an end-to-end processing framework for retinal prostheses, aligning with the domain of **applications to neuroscience & cognitive science** within ICLR. Comparable researches [1,2,3,4] to StimuSEE, have been acknowledged by NIPS and ICLR in recent years.
> >
> > **Q9: Not quite clear to me what “the front 20 times prediction of” mean in Figure 5 (b) caption. What are the x-axes “Times” in this plot? is it seconds?**
> >
> > **A9:** Please refer to **A7**. We will revise Fig. 5b in the camera-ready version for better presentation.
> >
> > **Q10: Given the four repeats in your V1 data, could you calculate the noise ceiling / self consistency of V1 response itself? then you can kind of compare your obtained pearson correlation to that “ceiling”.**
> >
> > **A10:** Visual neurons exhibit varying responses even to the same stimulus. In this study, we utilized the average value of these neural responses across four trials as the ground truth label. From this perspective, the average inherently provides a degree of self-consistency.
> >
> > **Q11: How do you compute the energy efficiency of Spiking RNN? also by counting the multiplication and addition in the simulated neuron firing process? —— if that’s the case won’t that depend on how do you discretize time?**
> >
> > **A11:** The energy efficiency is calculated based on the number of multiplication and addition operations. In this study, a 33 ms output from a DVS sensor with a 1 $\mu s$ time resolution serves as input to the Spiking RNN.
> >
> > **Q12: For more data to train the model, [5] seems to be quite relevant.**
> >
> > **A12:** Thank you for sharing this reference. We will examine it closely and explore ways to enhance our future work by incorporating insights from [5].
> >
> >
> > # Reference
> >
> > [1] Granley J, Relic L, Beyeler M. Hybrid neural autoencoders for stimulus encoding in visual and other sensory neuroprostheses[J]. Advances in Neural Information Processing Systems, 2022, 35: 22671-22685.
> >
> > [2] Konstantin-Klemens Lurz, Andreas S. Tolias, and Fabian H. Sinz. Generalization in data-driven models of primary visual cortex. International Conference on Learning Representations, 2021.
> >
> > [3] Bashiri M, Walker E, Lurz K K, et al. A flow-based latent state generative model of neural population responses to natural images[J]. Advances in Neural Information Processing Systems, 2021, 34: 15801-15815.
> >
> > [4] Lindsey J, Ocko S A, Ganguli S, et al. A unified theory of early visual representations from retina to cortex through anatomically constrained deep CNNs[J]. International Conference on Learning Representations, 2019.
> >
> > [5] Chen, X., Wang, F., Fernandez, E., & Roelfsema, P. R. (2020). Shape perception via a high-channel-count neuroprosthesis in monkey visual cortex. Science, 370(6521), 1191-1196.

---

### Meta-Review · Area_Chair_Cecn · 2023-12-09

**Metareview:**

The paper describes a retinal prosthesis system aimed to restore vision for human patients with impaired or degenerated retinae but functional retinal ganglion cells. The system includes a spike-based, time-continuous imager (DVS), followed by a spiking recurrent neural network and associated phosphene model to compute the proper stimulation pattern of the ganglion cells, followed by a convolutional based output network and spike generator that then predicts the firing patterns of neurons in primary visual cortex. The system (in particular the output network) is trained to optimally predict the real V1 neural firing patterns to the same visual stimulations. The authors refer to this as end-to-end optimization which is one of the key novel features of the proposed solution compared to current state-of-the-art systems.

The reviewers much appreciated this novel aspect of the work. They also valued the spike-based computation of the entire system (with the exception of the output network) that promises high energy efficiency, which is crucial for a potential clinical application. The potential of the latter was mentioned to be another plus of the work. Two main criticisms were raised: first, the description of the training methodology was unclear to the point that authors had to clarify what was trained how in the follow-up discussion with the reviewers. Second, pearson correlation coefficients as benchmark to compare between measured and predicted V1 neural responses was deemed to reflect a rather coarse metric.

The AC agrees in large part with both the described strengths and weaknesses of the work. The novelty of optimizing retinal stimulation patterns for neural responses in V1 rather than e.g. image reconstruction (as in previous work) seems definitely the way to go, and thus the approach promises large potential for clinical applications. Some of the technical details of the training and evaluation of the system show room for improvement. The descriptions of some of these details could also be clearer in particular with regard to which parameter and aspects of the processing pipeline are trained and if, how they were trained. It is unclear how many of the clarifications made during the discussions with the reviewers found their way into the final version of the manuscript.

**Justification For Why Not Higher Score:**

From a technical implementation point of view the proposed work relies on standard technologies, and thus does not offer that much for the ICLR community.

**Justification For Why Not Lower Score:**

The novelty of the overall approach is high and its potential for clinical applications is promising. This definitely makes it worth being presented.

---

### Decision · Program_Chairs · 2024-01-16

Accept (poster)